# Characterization and Assessment of 2, 4-Diacetylphloroglucinol (DAPG)-Producing *Pseudomonas fluorescens* VSMKU3054 for the Management of Tomato Bacterial Wilt Caused by *Ralstonia solanacearum*

**DOI:** 10.3390/microorganisms10081508

**Published:** 2022-07-26

**Authors:** Perumal Suresh, Murukesan Rekha, Subramanian Gomathinayagam, Vellaisamy Ramamoorthy, Mahaveer P. Sharma, Perumal Sakthivel, Karuppannan Sekar, Mariadhas Valan Arasu, Vellasamy Shanmugaiah

**Affiliations:** 1Department of Microbial Technology, School of Biological Sciences, Madurai Kamaraj University, Madurai 625 021, Tamil Nadu, India; sureshmicro90@gmail.com (P.S.); rekhakarishma@gmail.com (M.R.); 2Faculty of Agriculture and Forestry, University of Guyana, Berbice Campus, Tain, Georgetown P.O. Box 11110, Guyana; gomsrekha@uog.edu.gy; 3Department of Plant Pathology, Agricultural College and Research Institute, Tamil Nadu Agricultural University, Madurai 625 104, Tamil Nadu, India; rvrmoorthy@yahoo.com; 4Microbiology Section, Directorate of Soybean Research (DSR Indore-ICAR), Khandwa Road, Indore 452 001, Madhya Pradesh, India; mahaveer620@gmail.com; 5Department of Chemistry, Anna University—University College of Engineering, Dindigul 624 622, Tamil Nadu, India; sakthi30692@gmail.com (P.S.); karuppannansekar@gmail.com (K.S.); 6Department of Botany and Microbiology, College of Science, King Saud University, P.O. Box 2455, Riyadh 11451, Saudi Arabia; mvalanarasu@gmail.com

**Keywords:** *Pseudomonas fluorescens*, DAPG, bacterial wilt, reactive oxygen species, biocontrol efficacy

## Abstract

Microbial bio-products are becoming an appealing and viable alternative to chemical pesticides for effective management of crop diseases. These bio-products are known to have potential to minimize agrochemical applications without losing crop yield and also restore soil fertility and productivity. In this study, the inhibitory efficacy of 2,4-diacetylphloroglucinol (DAPG) produced by *Pseudomonas fluorescens* VSMKU3054 against *Ralstonia solanacearum* was assessed. Biochemical and functional characterization study revealed that *P. fluorescens* produced hydrogen cyanide (HCN), siderophore, indole acetic acid (IAA) and hydrolytic enzymes such as amylase, protease, cellulase and chitinase, and had the ability to solubilize phosphate. The presence of the key antimicrobial encoding gene in the biosynthesis of 2,4-diacetylphloroglucinol (DAPG) was identified by PCR. The maximum growth and antimicrobial activity of *P. fluorescens* was observed in king’s B medium at pH 7, 37 °C and 36 h of growth. Glucose and tryptone were found to be the most suitable carbon and nitrogen sources, respectively. DAPG was separated by silica column chromatography and identified by various methods such as UV-Vis, FT-IR, GC-MS and NMR spectroscopy. When *R. solanacearum* cells were exposed to DAPG at 90 µg/mL, the cell viability was decreased, reactive oxygen species (ROS) were increased and chromosomal DNA was damaged. Application of *P. fluorescens* and DAPG significantly reduced the bacterial wilt incidence. In addition, *P. fluorescens* was also found effective in promoting the growth of tomato seedlings. It is concluded that the indigenous isolate *P. fluorescens* VSMKU3054 could be used as a suitable biocontrol agent against bacterial wilt disease of tomato.

## 1. Introduction

*Ralstonia solanacearum* is a soil-borne plant pathogenic bacterium causing bacterial wilt disease in several crops, including tomato, worldwide and causing heavy economic loss [1]. Tomato bacterial wilt disease brings about nearly 60 to 100% crop loss [2,3]. The pathogen infects host plants by entering through natural wounds; it colonizes xylem tissues and blocks water channels in the xylem tissues, which results in a reduction in the supply of nutrients and water, leading to wilting in plants; finally, the host plant dies [4,5]. Its potent surviving capacity in various environments and wide host range make *R. solanacearum* very difficult to control using chemical pesticides [6]. Recent usage of chemical fertilizers and pesticides are hazardous to human and natural ecosystems. Thus, there is a need to find alternative methods in suppression of soil-borne phytopathogens to avoid the pollutants [7]. Thus, the application of potent anti-microbial agents producing strong inhibitory compounds is a promising alternative strategy.

Plant disease can be managed by biological control agents, especially *P. fluorescens* isolates, which efficiently control several bacterial and fungal plant diseases [8]. Garrido-Sanz et al. [9] demonstrated that the genus *Pseudomonas* includes 50 valid species and *P. fluorescens* is the most important bacterial species within the *Pseudomonas* group. *P. fluorescens* strains have been isolated from rhizosphere soil and shown to promote plant development. Hence, they are classified as plant growth promoting rhizobacteria (PGPR) [10,11]. They show direct and indirect mechanisms in suppression of plant pathogens and plant growth promoting activities [12,13,14,15]. Among the various mechanisms, the production of antimicrobial secondary metabolites such as phenazine, 2,4-diacetylphloroglucinol (2,4-DAPG), pyrrolnitrin, pyoluteorin, hydrogen cyanide and other metabolites is effective for the inhibition of soil-borne pathogens [16,17,18]. Among these metabolites, 2,4-DAPG is one of the most significant metabolites produced by *P. fluorescens* and it inhibits a broad spectrum of fungal and bacterial pathogens, including *R. solanacearum* [19,20,21]. It possesses anti-viral, antihelmintic and herbicidal activity along with other properties such as anti-lung, anti-leukemic and anti-breast cancer activity [22,23,24]. DAPG-producing *P. fluorescens* showed inhibitory effect against bacterial wilt in tomato. It significantly controlled bacterial wilt disease incidence under greenhouse conditions [21]. Several *Pseudomonas* species successfully control bacterial wilt disease caused by *R. solanacearum* in both in vitro and in vivo conditions [3,25,26,27]. Kwak et al. [28] reported the modes of action of DAPG, such as cleavage of the bacterial cell membrane, increasing ROS and DNA damage and finally leading to cell death. In this context, *P. fluorescens* and DAPG play a major role in controlling the bacterial wilt disease and increases the agricultural crop yield.

Keeping the importance of DAPG-producing *P. fluorescens* in suppression of various phytopathogens, the present study was carried out to characterize and assess the antagonistic activity and biocontrol efficacy of DAPG and DAPG-producing *P. fluorescens* VSMKU3054 against *R. solanacearum* and for the management of bacterial wilt in tomato.

## 2. Methods and Materials

### 2.1. Dual Culture Method

Antagonistic bacteria were isolated from tomato rhizosphere in Madurai district, Tamil Nadu, India. The antagonistic activity of *P. fluorescens* VSMKU3054 strain was tested against *R. solanacearum* and fungal phytopathogens viz. *Rhizoctonia solani, Sclerotium rolfsii, Macrophomina phaseolina* and *Fusarium oxysporum* by the dual culture plate technique. A dual culture plate assay was used to assess the antagonistic activity of *P. fluorescens* VSMKU3054 against *R. solanacearum* [29]. *R. solanacearum* (10^8^ CFU/mL) was seeded on nutrient agar (NA) medium (Adding bacterial culture to the medium and plating) and a loopful of *P. fluorescens* VSMKU3054 was center-point inoculated. After two days of incubation at 30 °C, the zone of inhibition was measured to determine antagonistic activity. Potato dextrose agar medium (PDA) (Himedia Laboratories, Mumbai, India) was used for the dual culture plate technique for testing antagonistic activity of *P. fluorescens* VSMKU3054 against fungal pathogens. The bacterial colony was streaked on one side of the Petri plate and the fungal disk was placed on the center of the plate. PDA plates inoculated with pathogen alone served as the control. Antagonistic activities were evaluated by measuring the inhibition zones after 2–7 days of incubation at 28 °C (when the pathogen covered the entire plate in the control treatment).

### 2.2. Assessing Carbohydrate Utilization and Hydrolytic Enzyme Production

The selected isolate VSMKU3054 was identified as *P. fluorescens* VSMKU3054 by 16S rDNA sequence analysis in our previous study [30]. The carbohydrate fermentation was tested using the Hicarbo™ kit (KB009) as described by Himedia Laboratories, Mumbai, India.

Protease production by the isolate VSMKU3054 was determined on skim milk agar. Similarly, amylase, gelatinase, cellulase and chitinase production by the isolate VSMKU3054 were determined on nutrient agar with respective substrates [31,32]. All the hydrolytic enzyme activities were identified by colour change or a clear zone formation around the colony.

### 2.3. Production of Antimicrobial Metabolites by VSMKU3054

Siderophore production by *P. fluorescens* VSMKU3054 was assessed by chrome azurol S (CAS) agar method [33]. A loopful of *P. fluorescens* was streaked onto CAS agar medium and three days after incubation at 30 °C, siderophore production was observed by a change of colour from blue to brownish orange in the medium.

One millilitre of *P. fluorescens* colony was streaked on King’s B plates and Whatman filter paper soaked in a solution containing 0.5% picric acid and 2% sodium carbonate was placed in the lid plate. The cultures were incubated for two to three days for the production of HCN. The colour change from yellow to brick red indicates HCN production [34].

A single colony of *P. fluorescens* VSMKU3054 was streaked on Pikovskaya’s agar medium for the production of phosphate solubilisation [35]. Three days after incubation at 30 °C, the clear zone was observed around the colony. Production of IAA by *P. fluorescens* VSMKU3054 was tested with slight modification [36]. *P. fluorescens* VSMKU3054 was inoculated with 0.4% L-tryptophan supplementation on King’s B broth and incubated for 24 h. The 2 mL of cell-free culture filtrate was mixed with 4 mL of Salkowski reagent after centrifugation at 10,000 rpm for 15 min (1 mL of 0.5 M FeCl_3_ in 50 mL of 35 percent perchloric acid). The mixture was incubated for 20 min in dark conditions at room temperature and the colour change was observed.

### 2.4. Detection of 2,4-DAPG Antimicrobial Encoding Genes in P. fluorescens by PCR

Detection of DAPG-encoding gene *phlD* (2,4-diacetylphloroglucinol) was performed by using gene specific primers viz., B2BF (5′-ACCCACCGCAGCATCGTTTATGAGC-3′) and BPR4 (5′-CCGGTATGGAAGATGAAAAAGTC-3′) [37]. The PCR reaction resulted in a 629-bp amplicon for DAPG. *P. fluorescens* CHAO was used as a positive control. PCR amplification was carried out (Smart PCR, Cyber Lab, Somerset, NJ, USA) with 25 µL reaction mixture containing DNA 30 ng, 10× PCR buffer, 10 μM dNTP’s, 10 pmol of forward and reverse primer and 1U of Taq DNA polymerase (GeNeT Bio, Nonsan, Korea). The PCR program consisted of an initial denaturation at 95 °C for 3 min, followed by 30 cycles at 94 °C for 60 s, 57.5 °C for 60 s, 72 °C for 60 s, and a final extension at 72 °C for 5 min. The amplified products were separated on 1% agarose gel (Seakem Lonza, Rockland, ME, USA) in 1× TBE buffer (Himedia Laboratories, Mumbai, India). Bands were visualized by ethidium bromide staining and viewed under a gel documentation system.

### 2.5. Optimizing Growth Conditions for the Production of Antimicrobial Metabolites by P. fluorescens VSMKU3054

Five culture media viz. nutrient broth (NB), King’s B medium (KBB), Luria–Bertani (LB), Tryptone soya broth (TSB) and nutrient sucrose broth (NSB) (Himedia Laboratories, Mumbai, India) were used for evaluating the production of secondary metabolites by *P. fluorescens* VSMKU3054. All the media were sterilized and inoculated with *P. fluorescens* and incubated for 24 h at 37 °C. The cultures were centrifuged at 10,000 rpm for 10 min. LB medium was seeded with *R. solanacearum* and poured on Petri dishes. Wells were formed in LB medium. Culture filtrates of various media were applied in the agar well. The inhibitory effect of culture filtrate of *P. fluorescens* against *R. solanacearum* was assessed by measuring the inhibition zone [23,38]. To assess the effect of temperature on antimicrobial metabolite production, *P. fluorescens* VSMKU3054 was inoculated in King’s B broth and grown at different temperatures from 20, 25, 28, 31, 34, 37 and 40 °C for 24 h in a shaker at 120 rpm. After incubation, the inhibitory effect of cell-free culture filtrate against *R. solanacearum* was assessed on LB agar medium by measuring the inhibition zone as explained above [23]. To evaluate the effect of pH conditions on the production of antifungal metabolites, King’s B medium was prepared with different pH levels from 4 to 10 by adjusting with 0.1 N NaOH or 0.1 N HCl (Himedia Laboratories, Mumbai, India). The medium was inoculated with *P. fluorescens* VSMKU3054 and incubated for 24 h in a shaker at 120 rpm at 37 °C. The inhibitory effect of cell-free culture filtrate activity against *R. solanacearum* was examined as described above [23].

To evaluate the effect of various carbon sources on the production of antifungal metabolites, King’s B medium was prepared with various carbon sources (glucose, glycerol, fructose, cellulose, sorbitol, sucrose, starch, arabinose, lactose and xylose) amended at a concentration of 1% for the production of antimicrobial secondary metabolites. To determine the effect of various nitrogen sources for the production of antimicrobial secondary metabolites, King’s B medium was prepared with various nitrogen sources viz., beef extract, potassium nitrate, arginine, ammonium sulphate, ammonium nitrate, ammonium chloride, urea, yeast extract, tryptone and sodium nitrate at concentrations of 1% and inoculated with *P. fluorescens* and incubated for 24 h. Inhibitory activity of cell-free culture filtrate against *R. solanacearum* was evaluated as described above. The various growth periods (incubation time) were evaluated for the production of metabolites with different time intervals by culturing *P. fluorescens* in King’s B medium incubated for 96 h. Production of metabolites was evaluated at every 12 h interval and the inhibitory activity was tested against *R. solanacearum* as described above.

### 2.6. Mass Production and Extraction of DAPG

The optimized condition was applied for DAPG production. A total of 7 L of bacterial culture medium was used for production of metabolites. For extraction of metabolites, culture supernatant was mixed with an equivalent volume of ethyl acetate in a separating funnel and extracts were dried using a rotary vacuum evaporator.

### 2.7. Purification and Characterization of Metabolites

Extracted crude metabolites were fractionated using a column chromatographic technique to separate antimicrobial compounds. The glass column was used for separation and the size of the column was 25 cm height × 2 cm diameter. The column was thoroughly washed thrice with acetone solvent and dried. The glass column was sealed with non-absorbent cotton. Further, the metabolites were dissolved in ethyl acetate and mixed with silica gel (mesh size 60–120) to make a slurry preparation and air-dried. The column was tightly packed with silica gel and slurry was added then filled with sufficient ethyl acetate to avoid the column drying. The mixture of solvent systems was hexane: ethyl acetate with different ratios used for the purification process. Different ratios of solvents started from 100:0, 99:1, 98:2 up to 0:100%. The column was run and fractions were collected. Each fraction was run on TLC and visualized in a UV chamber; then, the same fraction was mixed in a single tube. A total of 126 fractions were collected and all the fractions were subjected to TLC.

Metabolites were separated on TLC silica gel plates (20 cm × 20 cm, aluminum oxide 60 F_254_, Merck, Darmstadt, Germany) using the solvent chloroform: methanol (9:1) [29] along with standard phloroglucinol (Sigma, Kawasaki, Japan) and R_f_ value was calculated. Then, 2,4-DAPG was analyzed using a UV-vis spectrophotometer with an absorbance range between 200 and 400 nm. An FT-IR spectrum was recorded for 400–4000 cm^−1^ and the crude sample was mixed with KBr (Shimadzu, Kyoto, Japan). Further, purified DAPG and standard DAPG were analysed using GC-MS. Conversion and selectivity were measured by an Agilent 7820A GC using an HP-5 column of 30 mm length and high pure nitrogen (99.999%) as the carrier gas with an FID detector. The initial oven temperature was fixed at 50 °C and the temperature was gradually increased up to 280 °C at a rate of 10 °C/min. The observed products in the final reaction mixture were confirmed with a 5890A model of the Agilent GC-MS instrument using an HP-5 column with 30 mm length and high pure helium (99.99%) gas as the carrier gas. Finally, DAPG was analysed by nuclear magnetic resonance (NMR). The ^1^H NMR and ^13^C NMR spectra were recorded on a Bruker Avance spectrometer using acetonitrile-*d3* as solvent. ^1^H NMR was measured at 300 MHz and ^13^C NMR was measured at 128 MHz with chemical shifts (δ) expressed as values in parts per million (ppm). The splitting patterns in ^1^H NMR spectra are reported as follows: s = singlet; d = doublet; br s = broad singlet; m = multiplet. NMR (^1^H & ^13^C NMR) analyses for identifying the structure were drawn by Chemdraw software for the prediction of structure.

### 2.8. Assay of Minimum Inhibitory Concentrations (MIC) of DAPG and Its Effect on Cell Viability

The inhibitory effect of DAPG compounds was tested towards *R. solanacearum* in different concentrations (5–120 µg/mL). Live/dead assay of *R. solanacearum* against the pure compounds was performed by high content screening (HCS). DAPG at 90 µg/mL and tetracycline at 90 µg/mL were used as a positive control. Treated *R. solanacearum* was incubated for 8 h. After incubation, the cells were stained with acridine orange for live cells and propidium iodide for dead cell imaging [39]. The treated cells were analysed using the HCS imaging technique on a Perkin Elmer Operetta at 40× magnification and the image analysis was done using Harmony 3.0 software. Live cell percentage was calculated according to Yang et al. [40] with some modifications.
Percentage of live cell = Total No. of cell (green & red) − No. of dead cell (red)/Total No. of cell (green & red) × 100(1)

### 2.9. Determination of Reactive Oxygen Species (ROS)

The bacterial intracellular ROS formation was detected by cell-permeable fluorescent dye 2,7-dichlorofluorescein diacetate (DCFDA), which has a fluorescence wavelength of emission at 523 nm and excitation at 503 nm [41,42]. *R. solanacearum* was grown overnight and treated with DAPG at MIC (90 µg/mL) concentration. Further, it was incubated for 6 h. After incubation, *R. solanacearum* cells were centrifuged for 15 min at 4000 rpm and the pellet was resuspended with PBS. Harvested bacterial cells were washed three times and 10 mM of DCFDA was added. Further, cells were incubated for 20 min in dark. Then excess dye was removed by using PBS wash. Finally, the cells were analysed immediately with an HCS imaging technique at 40× magnification Perkin Elmer Operetta and the image analysis was done using Harmony 3.0 software. At the same time, the ROS production was determined in tetracycline-treated *R. solanacearum* used as a positive control and untreated cells considered as a negative control.

### 2.10. DNA Damage of R. solanacearum

The DNA-damaging effect of purified DAPG on *R. solanacearum* DNA was assessed by agarose gel electrophoresis [42]. Three different treatments such as DAPG (90 µg/mL), tetracycline and control (without treatment) were carried out against *R. solanacearum*. After treatment, DNA was isolated from bacteria by manual methods and run on 0.8% agarose gel and visualized by gel documentation.

### 2.11. Efficiency of P. fluorescens on the Suppression of R. solanacearum under Greenhouse Experiment

#### 2.11.1. Preparation of *P. fluorescens* and *R. solanacearum*

*P. fluorescens* VSMKU3054 was grown in KB medium for 24 h at 37 °C and the cells were centrifuged at 10,000 rpm for 10 min. After centrifugation, cells were washed with sterile phosphate buffer saline (PBS). Finally, they were resuspended in PBS. Further, bacterial cell suspensions were adjusted to 0.8 optical density at 600 nm (approximately 10^8^ CFU/mL). The *R. solanacearum* was grown overnight in TSB medium and the bacterial culture was centrifuged at 10,000 rpm for 10 min. Further, the pellets were resuspended in PBS and the cell suspensions were adjusted to the 0.8 OD at 600 nm used for this study.

#### 2.11.2. Preparation of Tomato Seedlings

PKM1 tomato seeds susceptible to bacterial wilt were obtained from the Horticultural College and Research Institute, Periyakulam, Tamil Nadu, India. A soil mixture consisting of soil: sand: vermicompost (2:1:1) was sterilized by autoclaving at 121 °C at 15 psi for 60 min. The potting soil mix was filled in clay pots (25 cm diameter; 8–10 kg capacity). The tomato seeds (PKM1) were surface sterilized using 1% sodium hypochlorite for 2 min and washed with sterile distilled water three times, then blot dried. Tomato seeds were grouped into three types such as control (sterile water), seeds treated with *P. fluorescens* culture and seeds treated with cell-free culture filtrate.

Thirty-day-old tomato seedlings (PKM1) were treated with *P. fluorescens* VSMKU3054 (10^8^ CFU/mL) with the following treatments. In the soil drenching method, after transplantation, 20 mL of *P. fluorescens* suspensions was poured into each pot. In the root dipping method, tomato roots were soaked in suspensions of *P. fluorescens* VSMKU3054 for 20 min and transplanted into pots. For the seed treatment, tomato seeds were soaked with *P. fluorescens* VSMKU3054 and sown in the pots (25 seeds/pot); seedlings were transplanted into new pots after 30 days of sowing. In the foliar spray method, *P. fluorescens* (20 mL) was sprayed on 30 days after transplantation. Similarly, cell-free culture filtrate (20 mL) was sprayed on 30-day-old transplanted tomato seedlings. In all the treatments, two tomato seedlings were transplanted per pot. All the treatments were conducted with three replications. All the pots were placed in a greenhouse at 28 °C and 80% relative humidity [43,44]. The pot culture experiment was laid out in greenhouse conditions using a completely randomized design.

T_1_—Control (without treatments).

T_2_—*R. solanacearum* alone.

T_3_—Tetracycline + *R. solanacearum.*

T_4_—*P. fluorescens* VSMKU3054 culture.

T_5_—*P. fluorescens* VSMKU3054 culture filtrate only.

T_6_—2,4-Diacetylphloroglucinol (DAPG) + *R. solanacearum* (Foliar Spray).

#### 2.11.3. Soil Application (SA)

T_7_—*P. fluorescens* VSMKU3054 culture + *R. solanacearum.*

T_8_—*P. fluorescens* VSMKU3054 culture filtrate + *R. solanacearum.*

#### 2.11.4. Root Dipping (RD)

T_9_—*P. fluorescens* VSMKU3054 culture + *R. solanacearum.*

T_10_—*P. fluorescens* VSMKU3054 culture filtrate + *R. solanacearum.*

#### 2.11.5. Seed Treatments (ST)

T_11_—*P. fluorescens* VSMKU3054 culture + *R. solanacearum.*

T_12_*—P. fluorescens* VSMKU3054 culture filtrate + *R. solanacearum.*

#### 2.11.6. Foliar Spray (FS)

T_13_—*P. fluorescens* VSMKU3054 culture + *R. solanacearum.*

T_14_—*P. fluorescens* VSMKU3054 culture filtrate + *R. solanacearum.*

After 10 days of transplanting, the soil was drenched with 20 mL of *R. solanacearum* (10^8^ CFU/mL) on the tomato root surface. In control, plants were treated with 20 mL of sterile water. In the positive control, 90 µg/mL (20 mL) of tetracycline was applied at the time of transplantation of tomato seedlings. Tomato plants were treated with DAPG at 90 µg/mL (20 mL). Afterward, the treated plants were monitored and disease incidence was observed and measured at 5–30 days. The disease incidence was recorded based on disease grade scale 0–4 with ranges: 0 = no wilting (healthy), 1 = 1–25%, 2 = 26–50%, 3 = 51–75% and 4 = 76–100% wilted or dead plant [43,45,46]. Finally, the biocontrol efficacy of *P. fluorescens* and compound were assessed. Fresh and dry weight and shoot and root lengths were measured at the 60th day. The biocontrol efficacy and disease incidence were also calculated.
Disease incidence = Sum of disease ratings/Total number of plants investigated × Highest disease index × 100%(2)
Biocontrol efficacy = Disease incidence in control × Disease incidence in treatments/Disease incidence in control × 100%(3)

### 2.12. Statistical Analysis

The greenhouse efficacy trial data were analysed using the analysis of variance (SAS Institute Inc., 1991, Cary, NC, USA). The data are mean values of three replicates ± standard deviation; means with different letters differ significantly at *p* = 0.05 according to Fisher’s LSD. The least significant difference (LSD) and Duncan’s Multiple Range Test (DMRT) were used to separate the treatment means.

## 3. Results

### 3.1. Antagonistic Activity of P. fluorescens VSMKU3054 against Plant Pathogens

The antimicrobial activity of *P. fluorescens* was investigated using a dual culture plate assay method against a variety of soil-borne plant pathogens. *P. fluorescens* exhibited 28.2 mm of inhibition zone against *R. solanacearum*; 24.57 mm against *R. solani*; 21.5 mm against *S. rolfsii*; 23.77 mm against *M. phaseolina*; and 27.13 mm against *F. oxysporum* (Table 1; Figure 1).

### 3.2. Phenotypic and Biochemical Characterization

The morphological identification of *P. fluorescens* VSMKU3054 under scanning electron microscopy (TESCAN VEGA3 SBH, Brno, Czech Republic) revealed that they are rod-shaped. It showed positive results for the catalase test, the oxidase test and the citrate utilization test, whereas the results were negative for the IMVIC test. Hicarbo test kit results showed that *P. fluorescens* utilized carbon sources such as lactose, galactose, maltose, fructose, trehalose, dextrose, melibiose, L-arabinose, glycerol, sucrose, mannose, mannitol, arabitol, D-arabinose, malonate and citrate, whereas it showed negative results for other carbon sources such as xylose, sorbitol, raffinose, inulin, sodium gluconate, salicin, inositol, dulcitol, alpha-methyl glucose, rhamnose, adonitol, melezitose, alpha-methyl D-mannose, cellobiose, ONPG, sarbose, xylitol and esulin.

### 3.3. Secondary Metabolites Produced from P. fluorescens VSMKU3054

P*. fluorescens* VSMKU3054 showed a positive result for hydrogen cyanide production, as revealed from colour change of the strip from yellow to brown (Figure 2C), and a positive result for siderophore production, as revealed from brown zone formation around the bacterial colony on CAS medium (Figure 2A). It also showed a positive result for phosphate solubilisation, as revealed from the formation of a clear zone around bacterial colonies (Figure 2D), and also produced the plant growth-promoting phytohormone indole acetic acid (Figure 2B).

### 3.4. Detection of DAPG Antibiotic Gene Amplification by PCR

*P. fluorescens* VSMKU3054 showed the presence of an antibiotic gene fragment for DAPG based on PCR analysis, as observed in *P. fluorescens* CHAO used as a positive control; the amplification of the DNA fragment was 629 bp. (Figure 3).

### 3.5. Optimizing the Growth Conditions for Antimicrobial Metabolite Production by P. fluorescens

The biomass production of *P. fluorescens* increased in five different culture media. Among the culture media tested, King’s B broth showed the highest production of biomass followed by NB, TSB and NSB in 2 days of incubation. The antimicrobial metabolite production was the maximum in KB medium, as revealed from the highest inhibition zone of 23.4 mm at 100 µL of sample (Figure 4A). With regard to temperature conditions, the maximum antimicrobial metabolites production by *P. fluorescens* VSMKU3054 was observed when it was grown at 37 °C, as revealed from the highest inhibition zone (22.33 ± 1.52 mm). Minimum antimicrobial activity (3 mm inhibition zone) was observed when *P. fluorescens* was grown at 40 °C (Figure 4C). With regard to pH conditions, the highest antimicrobial activity (24 mm inhibition zone) was noticed when *P. fluorescens* was grown at pH 7; the lowest antimicrobial activity was recorded at pH 10. (Figure 4B).

The influence of diverse carbon sources on antibacterial compound synthesis was investigated. The best carbon source for the formation of antibacterial compounds was found to be glucose, followed by sucrose, starch, lactose, sorbitol, glycerol and fructose. The least antibacterial activity was noticed when xylose was used as a carbon source (Figure 4D). Among the various nitrogen sources tested, tryptone supplementation supported the maximum antimicrobial metabolite production by *P. fluorescens* VSMKU3054, followed by beef extract, L-arginine and yeast extract, whereas urea and ammonium nitrate showed the minimum antimicrobial compound production (Figure 4E).

Among the various growth periods tested, culturing *P. fluorescens* for a period of 36–48 h resulted in the maximum production of antimicrobial bioactive metabolites, followed by 24, 12 and 60 h. The minimum production of bioactive metabolites was recorded at 84–96 h (Figure 4F).

**Figure 4 microorganisms-10-01508-f004:**
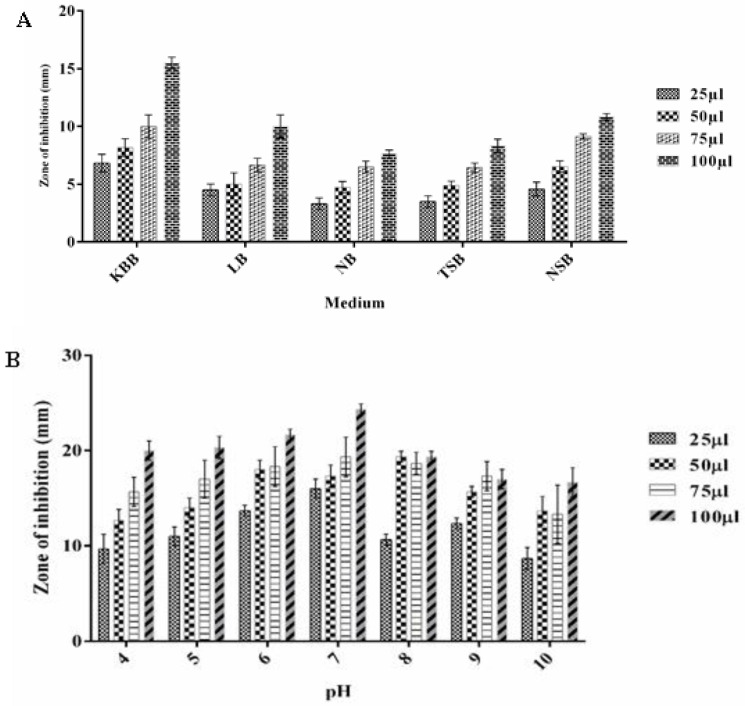
Optimization of *P. fluorescens* VSMKU3054 biomass with various abiotic parameters. (**A**) Medium, (**B**) temperature, (**C**) pH, (**D**) carbon sources, (**E**) nitrogen sources and (**F**) time.

### 3.6. Extraction and Purification of Secondary Metabolites

Purification of antimicrobial compounds from crude metabolites was separated by a silica column chromatographic technique. A total of 126 fractions were collected using column chromatography and all the fractions were run into TLC. After TLC analysis, sample fractions with the same retention time spot were mixed and pooled in the same tube. A total of nine compounds were selected and among the nine compounds, one compound numbered as 7 that matched the Rf value of standard phloroglucinol was collected and purified; the total amount of purified compound was 184 mg.

### 3.7. Characterization of Purified Compound

The purified compound 7, separated by TLC, has an Rf value of 0.82 that corresponds with standard phloroglucinol (Figure 5A). The compound 7 was analysed by a UV–vis spectrophotometer between the wavelengths of 200–600 nm and absorption peaks were obtained at 270 nm (Figure 5B). The FT-IR analysis results showed the peaks of 3473.8, 2981.95, 1757.15, 1608.7, 1458.18, 1244.09, 1055.06 and 929.69/cm and the functional group present in the FT-IR spectrum indicated the presence of phenolic OH groups, methyl groups, aromatic rings, and C–H groups in C–CH_3_ and aryl carbonyl compounds, and C–OH in alcohols, ethers, acid esters stretching, respectively (Table 2 and Figure 5C). *P. fluorescens* VSMKU3054 purified compound identification by GC-MS analysis obtained a mass spectrum at 210.4 *m*/*z*; further, it fragmented at 191.2, 175.1 and 147.1 m/z with a retention time of 10.893 min corresponding to authentic DAPG for further confirmation (Figure 5D).

^1^H and ^13^C NMR spectra were recorded on a 300 MHz Bruker NMR spectrometer using the solvent CD_3_CN. In ^1^H NMR.; diacetyl phloroglucinol peaks showed 2.63 singlet at 6H acetyl moiety (2 COCH_3_); 5.79 (singlet, 1H, Ar-H) aromatic proton of benzene ring and singlet 1.94 acetonitrile solvent peaks appeared; the assignments of the peaks are very well matched with compound structure (Figure 5E). ^13^C NMR spectra were recorded on 128 MHz using acetonitrile-d3 as solvent; peaks appeared (1.32 & 110.34 ppm), and the compound peaks showed 28.58 ppm (2 CH_3_), 95.47 ppm (Ar-H), 104.64 ppm (2 C-COCH_3_), 167.08 (Car-OH), 169.59 ppm (2 Car-OH) and 204.70 ppm (2 C=O) (Figure 5F).

The DAPG compound is visually observed as yellow in colour and soluble in ethyl acetate, chloroform, water, acetonitrile and methanol. The molecular formula of the DAPG antibiotic is C_10_H_10_O_5_ and molecular weight of the DAPG compound is 210 (Figure 5G).

**Figure 5 microorganisms-10-01508-f005:**
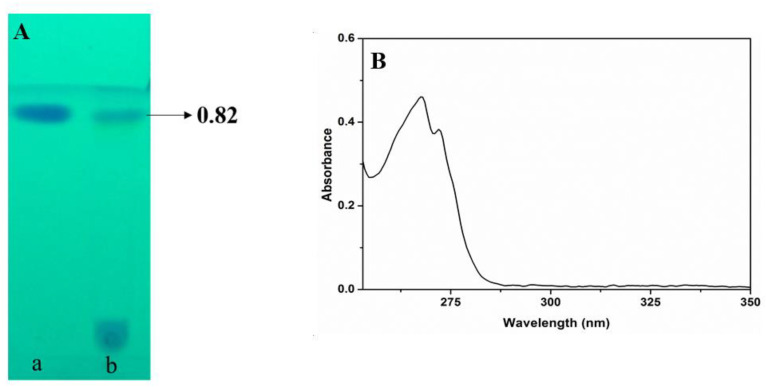
Characterization of 2,4-DAPG metabolites. (**A**) TLC; a: Authentic (Phloroglucinol). b: DAPG, (**B**) UV, (**C**) FT-IR, (**D**) GC-MS, (**E**) 1H, (**F**) 13C NMR and (**G**) structure of DAPG compound.

### 3.8. Bioactive Assay of DAPG against R. solanacearum

The purified DAPG compound isolated from *P. fluorescens* VSMKU3054 was tested to find its minimum inhibitory concentration against *R. solanacearum* with varying concentrations ranging from 5–120 µg/mL. The results showed that 90 µg/mL concentration of DAPG significantly inhibited *R. solanacearum* and was considered as the lowest concentration showing a greater zone of inhibition of 19.03 ± 0.35 mm. DAPG at 15 µg/mL showed no inhibition and antibacterial activity started from 20 µg/mL, showing a zone of inhibition when compared to commercial antibiotic tetracycline (Table 3 and Figure 6).

### 3.9. Live/Dead Cell Assay of R. solanacearum

Cells of *R. solanacearum* were treated with purified DAPG and they were observed under a high content screening imaging technique. Fluorescent dyes were used for visualizing intactness of cell membranes and nucleic acids. Cells with intact cell membranes showed green coloured cells, indicating live cells, and red coloured cells, considered as dead cells. Cells of *R. solanacearum* treated with DAPG showed a drastic reduction in viability compared to control. The DAPG treatment resulted in only 36% live cells and tetracycline (90 µg/mL) treatment resulted in 29% viable cells (Table 4; Figure 7).

### 3.10. Reactive Oxygen Species Assay—DAPG

A cell permeable assay was conducted by detection of ROS levels in *R. solanacearum* upon DAPG treatment using the fluorescent probe DCFDA. After treating *R. solanacearum* cells with DAPG, the generation of ROS accumulation in intracellular cell membranes was significantly increased, leading to cell membrane damage. The higher level of ROS accumulation was measured based on the increased intensity of the fluorescent probe that binds with both cell membranes and protein. Further, tetracycline treatment on *R. solanacearum* also significantly increased intensity of fluorescence bound to cell membranes and higher accumulation of ROS when compared to control. In control, there was no generation of ROS. The intracellular ROS accumulation was higher in tetracycline- than DAPG-treated cells (Figure 8).

### 3.11. DNA Damage Assay

DNA damage due to DAPG and tetracycline treatment of *R. solanacearum* was observed based on the intactness of total genomic DNA. Tetracycline-treated *R. solanacearum* caused damage and showed the low intensity of DNA, followed by DAPG-treated *R. solanacearum* cells, and DNA fragmentation was observed compared to control (Figure 9).

### 3.12. Biocontrol Efficiency of P. fluorescens VSMKU3054 and 2,4-DAPG against Tomato Bacterial Wilt Disease under Greenhouse Conditions

A total of 14 treatments were evaluated under greenhouse conditions along with control, and the efficacy of *P. fluorescens* VSMKU3054 and 2,4-DAPG against tomato bacterial wilt disease was assessed. Bacterial wilt common symptoms appeared in all the treatments challenge-inoculated with *R. solanacearum* from the fifth day’s post-inoculation (dpi). The observation of disease severity was calculated in all the treatments and the highest disease severity was observed in *R. solanacearum*-inoculated control plants (Figure 10).

All the treatments successfully reduced the severity of bacterial wilt disease. Among the various treatments, application of *P. fluorescens* as soil-drenching recorded the lowest percentage of disease incidence (46.6%) The wilt symptoms were delayed when the plants were applied either with *P. fluorescens* or tetracycline antibiotic. Further *P. fluorescens*-inoculated plants showed significantly increased root and shoot lengths, fresh and dry weights, and biomass (Table 5). *P. fluorescens* VSMKU3054 was observed to be the most effective treatment for application when compared with other treatments.

## 4. Discussion

In recent years, the utilization of microbes to improve plant growth and health has gained attention. Antibiotic production by soil saprophytic antagonistic organisms suppressed various phytopathogenic microbes in soil ecosystems [18,47]. Several methods have been developed to control bacterial wilt disease caused by *R. solanacearum*, including biological control, cultivation of resistant cultivators and soil organic amendments [48,49]. Biological control is one of the most prevalent and successful approaches for controlling plant diseases among these strategies. Several studies have documented that *P. fluorescens* is a significant biocontrol agent that suppresses soil-borne phytopathogens under invitro and in vivo conditions through direct or indirect mechanisms [3,10,11,50,51]. In the process of biological control, the selection of potential antagonistic microorganisms to suppress the growth of *R. solanacearum* and other soil-borne phytopathogens is considered as the first and foremost important step. Native isolates are more effective than exotic isolates in suppressing soil-borne diseases, according to numerous studies [25,52,53]. Previous studies have revealed that the selection of potential biocontrol agents was based on their antagonistic activity against *R. solanacearum* [49,54].

In this study, *P. fluorescens* VSMKU3054 showed the strongest antibacterial activity against *R. solanacearum* and antifungal activity against *R. solani*, *S. rolfsii*, *M. phaseolina* and *F. oxysporum* by significantly arresting mycelial growth with the production of antimicrobial metabolites. Similar to the results of the present study, earlier studies also reported that *P. fluorescens* strongly inhibited the growth of *R. solanacearum* by the production of different secondary metabolites [27,55]. Rai et al. [56] reported that *Pseudomonas* sp. strain RS-9 showed significant antagonistic activity against *R. solanacearum* and several fungal phytopathogens. *P. fluorescens* also showed strong antagonistic activity against fungal pathogens [36,57].

*P. fluorescens* produces potential antimicrobial compounds that are involved in enhancing plant growth promotion activity and also inhibition of plant diseases. In this study, *P. fluorescens* VSMKU3054 showed positive results for HCN, siderophore, phosphate solubilisation and IAA production. Furthermore, it produced lytic enzymes such as protease, amylase, chitinase, cellulose, pectinase and gelatinase, and improved plant growth as well as controlled plant disease. In accordance with the previous reports herein, *Pseudomonas* sp. produced antimicrobial compounds such as siderophore, HCN and hydrolytic enzymes that are implicated in fungal cell-wall degradation, reducing plant disease incidence and increasing plant growth promotion [44,52,58]. *P. fluorescens* VSMKU3054 harbours DAPG encoding gene fragments, which is confirmed by PCR analysis in accordance with the reports of Immanuel et al. [37].

Antibiotic production of fluorescent pseudomonas is affected by some abiotic factors such as carbon and nitrogen sources, temperature, pH and culture media [23,59,60]. *Pseudomonas* sp. produced 2,4-DAPG and culture conditions such as pH, temperature, and carbon and nitrogen sources influenced its total production [23,61]. Similarly, our study exhibited that metabolite production by *P. fluorescens* was influenced by abiotic factors for optimum growth. Accordingly, growth curve studies on various fluorescent pseudomonas indicated that maximum growth rate and lowest doubling time were observed in King’s B medium [62]. The present study also showed that the highest growth rate was noticed when *P. fluorescens* was cultured in King’s B medium.

Previous studies reported that fluorescent pseudomonas play an important role in producing antibiotics and thereby controlling phytopathogens [31,63]. DAPG possesses anti-helminthic, anti-viral and herbicidal activities, and it is also reported to be toxic against bacteria as well as fungi. *P. brassicacearum* J12 was identified with the 2,4-DAPG compound, which significantly inhibited *R. solanacearum.* DAPG shows broad-spectrum antimicrobial activity and it targets one or more cellular processes [23]. In our study, TLC, UV and FT-IR spectrophotometers confirmed that *P. fluorescens* VSMKU3054 produced the 2,4-DAPG compound, and the results were in accordance with previous reports of Ayyadurai et al., [64]. According to Brucker et al. [65] UV-visible spectrometer examination of DAPG revealed a peak of about 270 nm. In this investigation, GC-MS analysis revealed that the chromatogram peak of DAPG from *P. fluorescens* VSMKU3054 coincided with that obtained from standard DAPG [66,67]. Further confirmation of DAPG was done with ^1^H and ^13^C NMR and their structural elucidation of the compound was identified as DAPG, which coincided with previous literature data [66,67,68].

The antibacterial activity of various *Pseudomonas* sp. producing DAPG compounds was tested towards *R. solanacearum* and the results exhibited that the highest zone of inhibition was observed in isolate EC21 [25]. Nagendran et al. [69] reported that the effect of 2,4-DAPG produced by *P. fluorescens* TNAU PF1 strains towards the bacterial wilt pathogen *R. solanacearum* was tested by the agar well diffusion method, and the zone inhibition was observed to be 15.67 mm. In the present study, MIC was determined as 90 µg/mL against *R. solanacearum*. Live/dead assay (39%) of DAPG compounds produced by *P. fluorescens* VSMKU3054 strongly reduced the growth of *R. solanacearum* as compared to cells treated by tetracycline (29%) and untreated cells. The results of the present study were in accordance with the previous study, which showed that *R. solanacearum* had a lower number of live cells [39].

Oxygen is essential for all living organisms and it is also a precursor of ROS, due to damage caused to organelles such as nucleic acids, proteins and other components. ROS lead to the damage of cell membranes and increase lipid peroxidation, which can cause membrane leakage and deteriorating effects on the respiratory system, resulting in apoptosis [42,70,71]. In the present study, 2,4-DAPG treatment of *R. solanacearum* showed substantial DNA damage. Further, significant DNA fragmentation was observed at 90 µg/mL concentrations, which was concordant with the reports of Cai et al. [42] and Roy et al. [72].

The biocontrol efficiency of *P. fluorescens* VSMKU3054 significantly reduced wilt disease incidence in tomato plants caused by *R. solanacearum* compared to antibiotic control under in vivo conditions. Zhou et al. [21] previously reported that *P. fluorescens* having the *phlF* gene exhibited a higher zone of inhibition against *R. solanacearum*; further, 2,4-DAPG production was enhanced in *phlF-* mutant strains and superior colonization was observed in the rhizosphere, which might have effectively suppressed soil-borne pathogens. Furthermore, the wild-type J2 strain showed significant biocontrol efficacy against tomato bacterial wilt. The potential antagonistic bacterium *P. protegens* RS-9 strains could also produce 2,4-DAPG and successfully controlled bacterial wilt of tomato [56]. Recent studies showed that *B. velezensis* B63 and *P. brassicacearum* P142 isolates are promising biocontrol agents against bacterial wilt in tomato plants under field conditions due to the production of plant growth promoting traits and induction of plant defense enzymes to suppress soil-borne pathogens [27]. Qessaoui et al. [73] recently reported that *Pseudomonas* strain Q13B played an important role in plant growth promotion under greenhouse conditions by increasing plant length and collar diameter compared to control. In accordance with the previous studies, the results of the present study showed that *P. fluorescens* VSMKU3054 significantly suppressed the bacterial wilt incidence and further increased the shoot and root lengths of tomato plants. Furthermore, it could be used as a biofertilizer and biocontrol agent to replace alternative chemical pesticides, improve plant growth in tomato plants and inhibit soil-borne pathogens [3,48,74,75].

## 5. Conclusions

In conclusion, the present study suggests higher antagonistic activity of DAPG-producing *P. fluorescens* VSMKU3054 against *R. solanacearum* and plant growth-promoting activity. *P. fluorescens* VSMKU3054 has the ability to colonize plant systems and also might express superior biocontrol activity against tomato bacterial wilt disease in both in vitro and in vivo conditions. The study strongly recommends that *P. fluorescens* VSMKU3054 be used as a formulated biopesticide that substitutes for chemical pesticides and also reduces harmful effects on the environment.

## Figures and Tables

**Figure 1 microorganisms-10-01508-f001:**
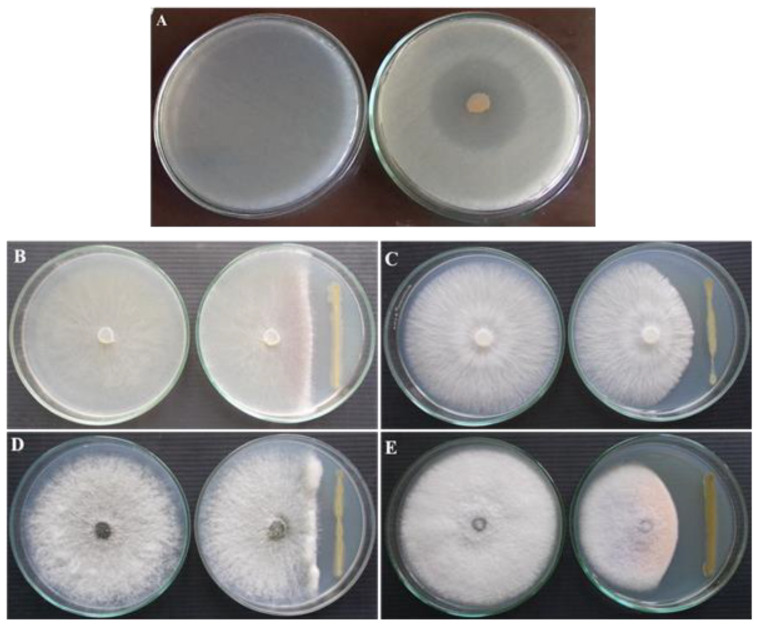
Antagonistic activity of *P. fluorescens* VSMKU3054 against plant pathogens by dual culture plate assay. (**A**)—*P. fluorescens* + *R. solanacearum*, (**B**)—*P. fluorescens* +*R. solani*, (**C**)—*P. fluorescens* + *S. rolfsii*, (**D**)—*P. fluorescens* +*M. phaseolina* and (**E**)—*P. fluorescens* + *F. oxysporum*.

**Figure 2 microorganisms-10-01508-f002:**
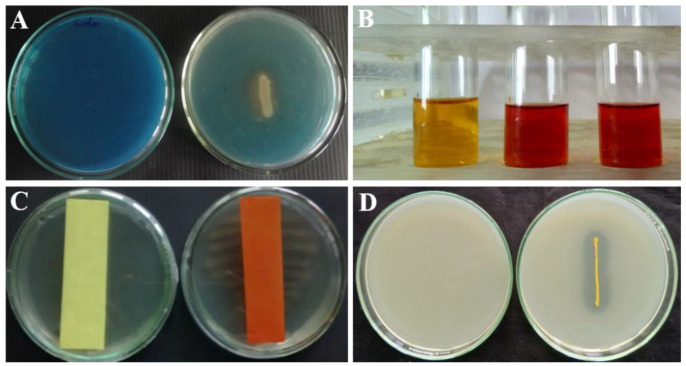
Secondary metabolites produced from *P. fluorescens* VSMKU3054. (**A**) siderophore, (**B**) indole acetic acid, (**C**) hydrogen cyanide & (**D**) phosphate solubilization.

**Figure 3 microorganisms-10-01508-f003:**
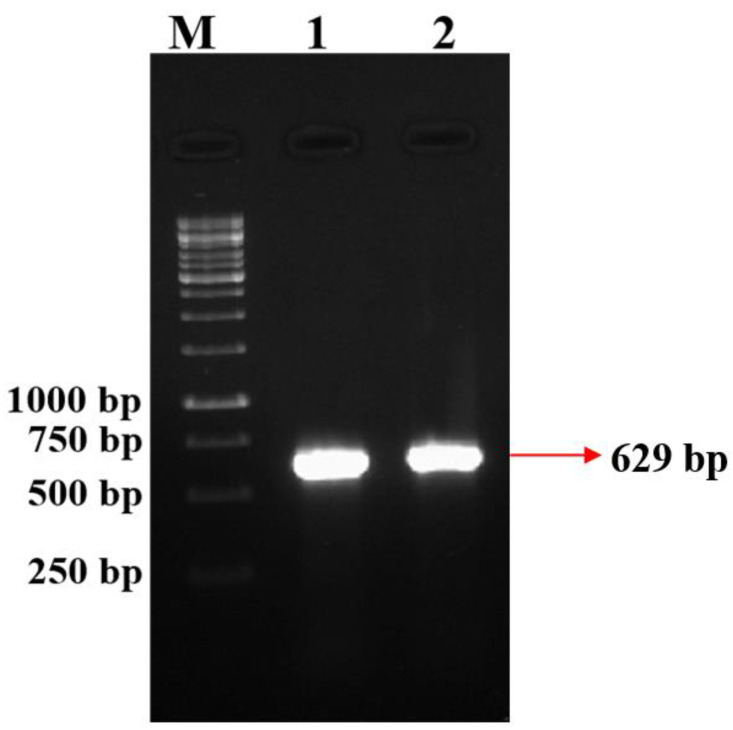
Detection of DAPG encoding gene from *P. fluorescens* VSMKU3054. M-Marker (1 kb), 1—CHAO, 2—*P. fluorescens*.

**Figure 6 microorganisms-10-01508-f006:**
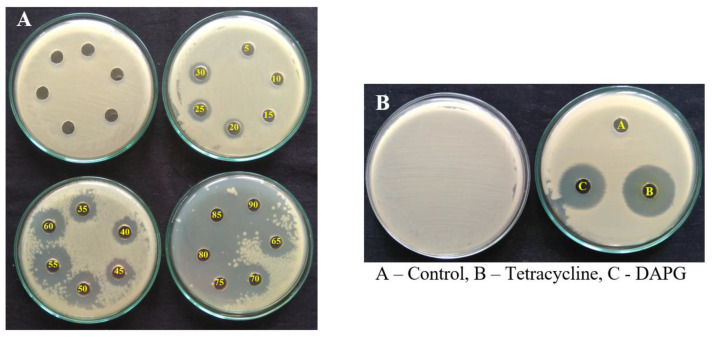
MIC of DAPG compound against *R. solanacearum*. (**A**) Different concentrations of DAPG metabolites against *R. solanacearum* and (**B**) 90 µg/mL concentration of tetracycline and DAPG.

**Figure 7 microorganisms-10-01508-f007:**
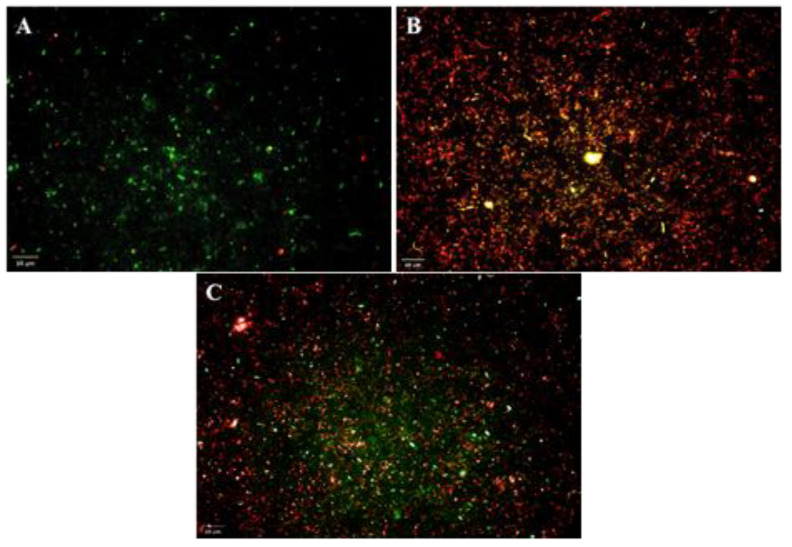
Live/dead assay of *R. solanacearum* in high content screening imaging system. Green and red stained cells considered as live and dead cells, respectively. (**A**) *R. solanacearum* alone (Control), (**B**) *R. solanacearum* treated with tetracycline at 90 µg/mL and (**C**)—*R. solanacearum* treated with DAPG at 90 µg/mL. Scale bar: 20 µm.

**Figure 8 microorganisms-10-01508-f008:**
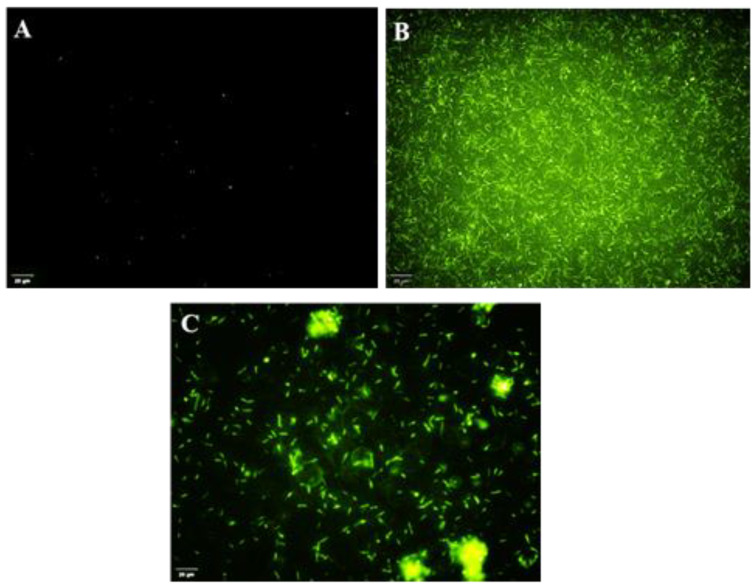
Accumulation of reactive oxygen species in *R. solanacearum* cells treated with DAPG at 90 µg/mL concentration at 6 h incubation time. Green fluorescence indicates the DCFDA dye binds with ROS accumulation in *R. solanacearum* cell membrane. (**A**) Control (untreated), (**B**) tetracycline and (**C**) DAPG. Scale bar: 20 µm.

**Figure 9 microorganisms-10-01508-f009:**
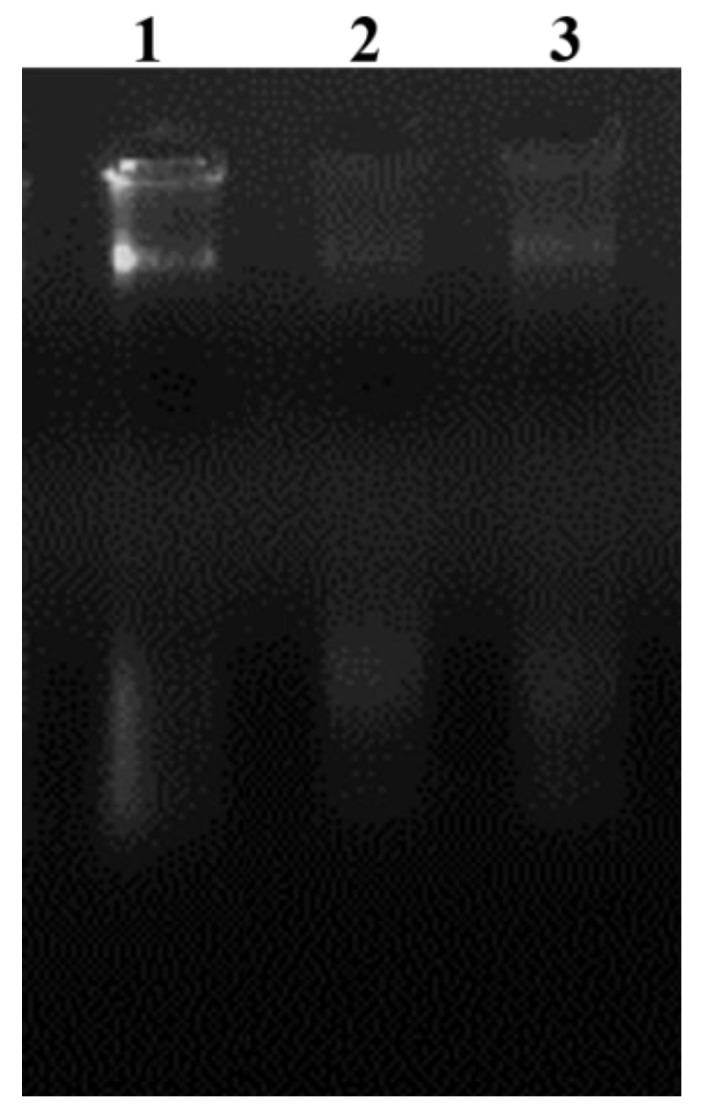
Agarose gel electrophoresis analysis of genomic DNA in *R. solanacearum* treated with DAPG and tetracycline. Lane 1: control; 2: tetracycline; and 3: DAPG.

**Figure 10 microorganisms-10-01508-f010:**
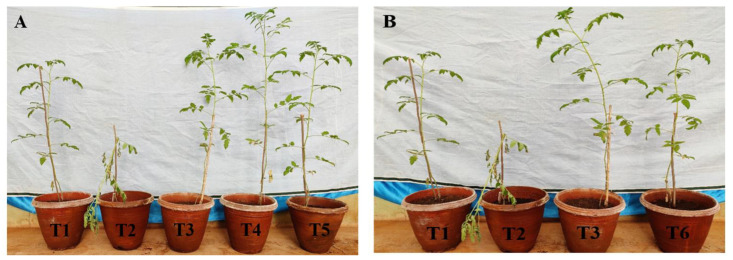
Biocontrol efficacy of *P. fluorescens* and 2,4-DAPG against bacterial wilt pathogen *R. solanacearum* in tomato plants under greenhouse conditions. (**A**) T1-control, T2-*R. solanacearum* alone, T3-Tetracycline, T4-*P. fluorescens* (Pf), T5-*P. fluorescens* culture filtrate (CF); (**B**) DAPG compound treated plants (T6); (**C**) soil application of Pf and CF (T7&T8); (**D**) root dipping method of Pf and CF (T9&T10); (**E**) seed treatments of Pf and CF (T11&T12); (**F**) foliar spray methods of Pf and CF (T13&T14) treated with tomato plants.

**Table 1 microorganisms-10-01508-t001:** Antagonistic activity of *P. fluorescens* VSMKU3054 against *R. solanacearum* and fungal plant pathogens.

S. No.	Plant Pathogens	Zone of Inhibition (mm)
1.	*R. solanacearum*	28.2 ± 0.6
2.	*R. solani*	24.57 ± 0.86
3.	*S. rolfsii*	21.5 ± 0.89
4.	*M. phaseolina*	23.77 ± 0.38
5.	*F. oxysporum*	27.13 ± 0.76

**Table 2 microorganisms-10-01508-t002:** Functional groups of purified DAPG by FT-IR analysis.

S. No.	Wavelength (cm^−1^)	Functional Groups
1.	3473.8	Phenolic OH group
2.	2981.95	Methyl group
3.	1757.15	Aromatic ring
4.	1608.7	C–H groups in C–CH_3_ compound
5.	1458.18	Aryl carbonyl compounds
6.	1244.09	C–OH in alcohols
7.	1055.06	Ethers
8.	929.69	Acid esters

**Table 3 microorganisms-10-01508-t003:** Minimum inhibitory concentration of DAPG from *P. fluorescens* VSMKU3054 against *R. solanacearum*.

S. No.	Concentration (µg/mL)	Zone of Inhibition (mm)	
		DAPG	Tetracycline
1.	5	0 ± 0	0 ± 0
2.	10	0 ± 0	0 ± 0
3.	15	0 ± 0	9.83 ± 1.04
4.	20	1.83 ± 0.76	11.63 ± 0.35
5.	25	2.33 ± 0.57	13.37 ± 0.71
6.	30	3.33 ± 0.57	15.27 ± 0.7
7.	35	5.67 ± 1.04	13.6 ± 0.4
8.	40	7.67 ± 0.57	14.7 ± 0.31
9.	45	7.83 ± 0.49	14.9 ± 0.11
10.	50	8.4 ± 0.5	15.3 ± 0.6
11.	55	10 ± 0.4	15.67 ± 0.58
12.	60	10.9 ± 0.35	16.13 ± 0.23
13.	65	13.4 ± 0.61	16.73 ± 0.25
14.	70	15.53 ± 0.41	17.13 ± 0.23
15.	75	16.67 ± 1.15	18.83 ± 0.29
16.	80	17.57 ± 0.4	19. 67 ± 1.15
17.	85	18.43 ± 0.55	20.5 ± 0.5
18.	90	19.03 ± 0.35	21 ± 1
19.	95	20.17 ± 1.04	22.83 ± 1.04
20.	100	20.67 ± 0.58	23 ± 1
21.	110	21.5 ± 0.5	24.33 ± 0.28
22.	115	22.33 ± 0.57	24 ± 1
23.	120	22.5 ± 0.86	25.5 ± 0.5

**Table 4 microorganisms-10-01508-t004:** Live/dead assay of *R. solanacearum*.

S. No.	Treatments	Percentage of Live Cells	No. of Fields
1.	*R. solanacearum*	89	9
2.	*R. solanacearum* + Tetracycline	29	9
3.	*R. solanacearum* + DAPG	36	9

**Table 5 microorganisms-10-01508-t005:** Biocontrol efficiency of *P. fluorescens* VSMKU3054 and 2,4-DAPG against bacterial wilt disease in tomato (PKM1) growth under greenhouse conditions.

Treatments	Shoot Length (cm)	Root Length (cm)	No. of Branches	Fresh Weight (g)	Dry Weight (g)	Disease Incidence (%)	Biocontrol Efficacy (%)	Biomass Increase (%)
T_1_ (Control)	93.33 ^cd^ ± 1.52	25.67 ^f^ ± 2.08	12.0 ^ef^ ± 1	20.23 ^d^ ± 1.06	3.36 ^cde^ ± 0.45	0 ± 0	0 ± 0	35.45 ^ef^ ± 1.11
T_2_ (*R. solanacearum*)	79.0 ^f^ ± 1	15.0 ^g^ ± 1.73	11.67 ^f^ ± 1.52	15.08 ^e^ ± 0.68	2.31 ^f^ ± 0.18	94.97 ± 1.87	0 ± 0	0.0 ^h^ ± 0.00
T_3_ (Antibiotic + R.S)	96.67 ^c^ ± 1.15	31.33 ^e^ ± 2.51	14.33 ^cde^ ± 1.15	24.7 ^b^ ± 1.35	4.32 ^b^ ± 0.27	23.70 ± 0.45	75.56 ± 0.88	63.76 ^b^ ± 3.61
T_4_ (*P. fluorescens* culture) (SA)	112.67 ^a^ ± 2.08	53.0 ^a^ ± 2.64	22.33 ^a^ ± 1.52	27.72 ^a^ ± 1.58	5.40 ^a^ ± 0.21	0 ± 0	0 ± 0	80.65 ^a^ ± 7.87
T_5_ (*P. fluorescens* culture filtrate) (SA)	104 ^b^ ± 1	34.0 ^cde^ ± 3.60	12.0 ^ef^ ± 1	22.56 ^bcd^ ± 1.3	4.34 ^b^ ± 0.26	0 ± 0	0 ± 0	49.56 ^c^ ± 2.19
T_6_ (Compound + R.S) (FS)	104.33 ^b^ ± 2.51	22.0 ^f^ ± 2.64	15.67 ^bc^±1.52	21.77^d^ ± 2.12	4.23 ^b^ ± 0.1	47.61 ± 1.03	49.86 ± 0.23	47.54 ^cd^ ± 3.13
Soil Drenching								
T_7_ (Culture + R.S)	106.0 ^b^ ± 2	31.67 ^de^±2.51	17.0 ^b^±1.73	22.1 ^cd^ ± 1.53	4.43 ^b^ ± 0.12	46.66 ± 1.44	50.85±1.48	46.61 ^cd^ ±9.28
T_8_ (Culture filtrate + R.S)	86.33 ^e^ ± 1.15	23.33 ^f^ ± 2.08	16.33^bc^ ± 0.57	20.06 ^d^ ± 1.07	3.22 ^de^ ± 0.19	70.06 ± 2.08	26.21 ± 2.12	29.99 ^f^ ± 6.18
Root Dipping								
T_9_ (Culture + R.S)	92.33 ^d^ ± 1.15	40.0 ^b^ ± 1.00	21.67 ^a^ ± 1.52	24.31 ^bc^ ± 1.24	3.28 ^de^ ± 0.38	69.73 ± 2.54	26.56 ± 2.71	61.27 ^b^ ± 5.24
T_10_ (Culture filtrate + R.S)	72.0 ^g^ ± 3	25.0 ^c^ ± 2.00	13.0 ^def^ ± 1	15.83 ^e^ ± 1.08	2.98 ^e^ ± 0.19	71.01 ± 1.34	25.18 ± 0.32	16.38 ^g^ ± 1.65
Seed Treatment								
T_11_ (Culture + R.S)	83.67 ^e^ ± 1.52	24.33 ^f^ ± 1.52	14.33 ^cde^ ± 0.57	21.46 ^d^ ± 0.52	3.26 ^de^ ± 0.18	46.75 ± 1.32	50.76 ± 1.32	42.42^cde^ ± 3.41
T_12_ (Culture filtrate + R.S)	104.33 ^b^ ± 3.21	35.67 ^cd^ ± 1.1	15.33 ^bcd^ ± 2.08	22.42 ^bcd^ ± 2.1	3.79 ^c^ ± 0.29	70.83 ± 1.36	25.41 ± 0.72	48.50 ^cd^ ± 7.91
Foliar Spray								
T_13_ (Culture + R.S)	95.33 ^cd^ ± 1.52	31.67 ^de^ ± 2.08	12.33 ^ef^ ± 0.57	21.05 ^d^ ± 1.22	3.70 ^cd^ ± 0.24	70.1 ± 2.02	26.17 ± 2.05	39.60 ^de^ ± 5.58
T_14_ (Culture filtrate + R.S)	83.67 ^e^ ± 2.51	36.0 ^c^ ± 2.04	17.67 ^b^ ± 1.52	21.47 ^d^ ± 0.85	3.53 ^cd^ ± 0.21	48.74 ± 2.63	48.68 ± 2.28	41.54 ^cde^ ± 4.38
LSD (0.05)	3.41	3.77	2.20	2.25	0.43	-	-	8.60

Data are average of 3 replications; LSD, Least significant difference (*p* = 0.05); The treatment means followed by same letter did not differ significantly by DMRT.

## Data Availability

Not applicable.

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
