# Peer review of "Characterization and Assessment of 2, 4-Diacetylphloroglucinol (DAPG)-Producing Pseudomonas fluorescens VSMKU3054 for the Management of Tomato Bacterial Wilt Caused by Ralstonia solanacearum"

_microorganisms, 2022, doi:10.3390/microorganisms10081508_

Round 1
Reviewer 1 Report
There are several commnets on the manuscript, please follow them. Please insert information about the replicates and/or number of samples in each experiments.

Author Response
Dear Sir
We have done as per your queries and give answers to all the questions.

Reviewer 2 Report
The authors have written a well-structured and thought article. The authors did a lot of work of experimental design, data analysis, and the results presenting. However, some revisions are still needed before accepting:
- In Celsius degrees author sometimes used ºC with a dash below º , sometimes oC - should be °C,
- Please provide some additional information about the source of reagents - There is no information about: names of reagents suppliers (e.g., agarose gel, TBE buffer, NaOH, HCl etc.), agars and medias suppliers (e.g. nutrient agar, potato dextrose agar, milk agar etc.),
- Please specify the strain and supplier of Rhizoctonia solani, Sclerotium rolfsii, Macrophomina phaseolina and Fusarium oxysporum,
- Figure 4 (C, D, E and F) is hardly visible, please improve their quality,
- Page 8 – please provide the name and supplier of scanning electron microscope.
Author Response
Dear Sir
We replied all the queries raised by you this modified manuscript
